# Microbial growth and carbon use efficiency show seasonal responses in a multifactorial climate change experiment

Eva Simon [1], Alberto Canarini [1✉], Victoria Martin[1], Joana Séneca [1], Theresa Böckle[1], David Reinthaler [2], Erich M. Pötsch[3], Hans-Peter Piepho[4], Michael Bahn[2], Wolfgang Wanek [1] & Andreas Richter [1✉]

Microbial growth and carbon use efficiency (CUE) are central to the global carbon cycle, as microbial remains form soil organic matter. We investigated how future global changes may affect soil microbial growth, respiration, and CUE. We aimed to elucidate the soil microbial response to multiple climate change drivers across the growing season and whether effects of multiple global change drivers on soil microbial physiology are additive or interactive. We measured soil microbial growth, CUE, and respiration at three time points in a field experiment combining three levels of temperature and atmospheric $CO_2$, and a summer drought. Here we show that climate change-driven effects on soil microbial physiology are interactive and season-specific, while the coupled response of growth and respiration lead to stable microbial CUE (average CUE = 0.39). These results suggest that future research should focus on microbial growth across different seasons to understand and predict effects of global changes on soil carbon dynamics.

[1] Terrestrial Ecosystem Research, Centre for Microbiology and Environmental Systems Science, University of Vienna, Vienna, Austria. [2] Department of Ecology, University of Innsbruck, Innsbruck, Austria. [3] Institute of Plant Production and Cultural Landscape, Agricultural Research and Education Centre, Raumberg-Gumpenstein, Austria. [4] Institute of Crop Science, University of Hohenheim, Stuttgart, Germany. ✉email: alberto.canarini@hotmail.it; andreas.richter@univie.ac.at

All organisms live and die. To live, grow and replicate, heterotrophic organisms need to assimilate organic carbon (C), and thus decompose organic material, that is provided upon the death of other organisms such as plants[1]. While most plant litter, the basis of the heterotrophic food web, is recycled back to the atmosphere as $CO_2$, a fraction enters the soil as microbial necromass, comprising the dead residues of soil microorganisms, which make up a conspicuous proportion of soil organic matter[2,3]. Over millennia, soil organic matter has accumulated to an amount of carbon exceeding that of the atmosphere and biosphere combined[4]. Thus, the processes that lead to the decomposition and accumulation of organic matter in terrestrial environments are both driven by the growth of heterotrophic microbial communities and their energy requirements. The efficiency by which microorganisms allocate carbon taken up to growth is termed microbial carbon use efficiency (CUE)[5-7]. Operationally, carbon use efficiency is usually defined as growth (i.e., new biomass production) over the sum of $CO_2$ production (mainly from respiration) and growth, as a proxy for uptake. This definition is a simplified view of microbial CUE, which reflects our current methodological limitations. Conceptionally, microbial CUE is determined by the balance between anabolic and catabolic processes in a cell[5,9], but current methods do not allow to account for biosynthetic processes, in which carbon is exuded, e.g., production of extracellular enzymes and metabolites, such as short-chain fatty acids[5,8,9]. While it is widely recognized that microbial physiology and community composition are strongly affected by extrinsic factors such as temperature, water availability, and supply of recent plant-derived carbon[6,10-12], the response of growth and CUE of soil microbial communities to global change drivers are not yet fully resolved. This is due to, amongst other reasons, the scarcity of studies assessing microbial physiology in field-based long-term climate change experiments.

Global climate change alters ecosystem carbon dynamics by concurrently modifying temperature, atmospheric $CO_2$ concentrations and water availability. These factors can have both direct and indirect effects on soil microbial physiology[13,14]. However, though climate projections predict coupled changes in these environmental factors[15], interactive effects of warming, elevated $CO_2$, and changes in precipitation are rarely considered[16-18]. This is one of the reasons, why the terrestrial carbon cycle remains the least constrained component of the global carbon cycle[19,20], especially when modelling the effects of multiple global climatic change drivers. A range of studies showed that elevated temperature had a stimulating effect on microbial activity, as enzymatic reactions are generally temperature sensitive[14,21-23]. Reaction rates increase to a certain temperature, referred to as optimal temperature, beyond which they decline again[21,24]. The same applies to the growth of microorganisms. The thermal optimum of microbial communities however shifts seasonally with changing soil temperature[25]. Drought can also directly affect microorganisms[26,27]. In order to maintain their intracellular water potential and to prevent cell damage, microorganisms need to synthesize organic osmolytes. This is costly both in terms of carbon and energy and might slow down or even stop microbial growth[26], while respiratory processes for maintenance are preserved[6]. In contrast to warming and drought, elevated atmospheric $CO_2$ concentrations only indirectly affect soil microorganisms. $CO_2$ fumigation was found to increase plant biomass production, at least initially[28-31], and to promote higher plant belowground carbon allocation[28,31-33]. Higher root biomass and activity possibly cause increased carbon availability to soil microbes[34,35]. Furthermore, elevated atmospheric $CO_2$ was observed to lead to improved plant water use efficiency[30,31,36-38], thus resulting in enhanced soil water availability for microbes.

Field studies investigating climate change effects on microbial growth and CUE are scarce and have mostly focused on effects of warming. Studies on the response of microbial CUE to warming reported contradictory findings: some authors reported no effects of warming on microbial CUE[39,40], while others observed reduced CUE[5-7,34,41-45], or increased microbial CUE[12]. However, these studies used a range of different approaches to estimate CUE[5], which may not allow direct comparisons[6,8,46]. In addition, many of these studies have been conducted with soils sampled from various ecosystems (forest, grassland, and agricultural land) and in different seasons or from experiments with different durations. Indeed, seasonal variations in environmental factors and in plant carbon inputs result in a recurrent change of the active microbial community throughout the year[14,47-49]. Depending on the time of the year, different factors may control microbial activity[22]. For example, during the cold seasons, temperature is considered a major limiting factor, whereas water availability might play a much bigger role in constraining microbial activity during summer[22]. Yet, seasonal dynamics are often ignored in climate change studies, although it was demonstrated that they can strongly modulate the response of grasslands to climate change[36]. More generally, microbial physiology (aside of microbial respiration or soil enzyme activities) has rarely been studied in multifactorial climate change experiments, restricting our predictive power of how the balance between anabolic and catabolic microbial processes, and thus the potential to store soil carbon, will change in a future climate[16,17].

To investigate interactive effects of multiple global change factors (warming, elevated $CO_2$, and drought) and their seasonal dynamics, we collected soil samples at three different time points during the growing season from a multifactorial climate change experiment in a sub-montane managed grassland[50], in which the treatments have been applied for four consecutive years. We had two main questions, namely, (i) how single or combined climate change drivers affect microbial growth, respiration, and CUE across seasons (within the growing season) and (ii) whether the effects of multiple climate change drivers on microbial physiology are additive or interactive.

In order to answer these questions, combinations of three levels of temperature (ambient, 1.5 °C and 3 °C above ambient temperature) and three levels of atmospheric $CO_2$ (ambient, 150 ppm and 300 ppm above ambient) were established in a surface response design. Drought was additionally superimposed on a subset of plots, i.e., on half of the high $CO_2$/high temperature treatment plots (future climate change scenario) and ambient climate plots, by the operation of automatic rain-out shelters in June and July. Drought plots were subjected to rewetting at the end of July, mimicking a strong precipitation event. Microbial growth was determined using the $^{18}O$ technique, which is based on measuring the incorporation of $^{18}O$ from water into genomic DNA. We hypothesized (i) that elevated temperatures and $CO_2$ alone would lead to increased microbial activity, while drought would generally reduce it, (ii) that the combined effects of these factors would be non-additive, and (iii) that the responses of microbial growth, respiration and CUE to climate change treatments would differ across seasons.

## Results

**Treatment effects on soil parameters and microbial biomass carbon.** Warming, elevated $CO_2$ and drought all affected soil moisture. Reductions in soil water content in warmed plots were evident throughout the growing season (Supplementary Fig. S1). Elevated $CO_2$ had a small positive effect on soil moisture. Exclusion of precipitation through rain-out shelters strongly

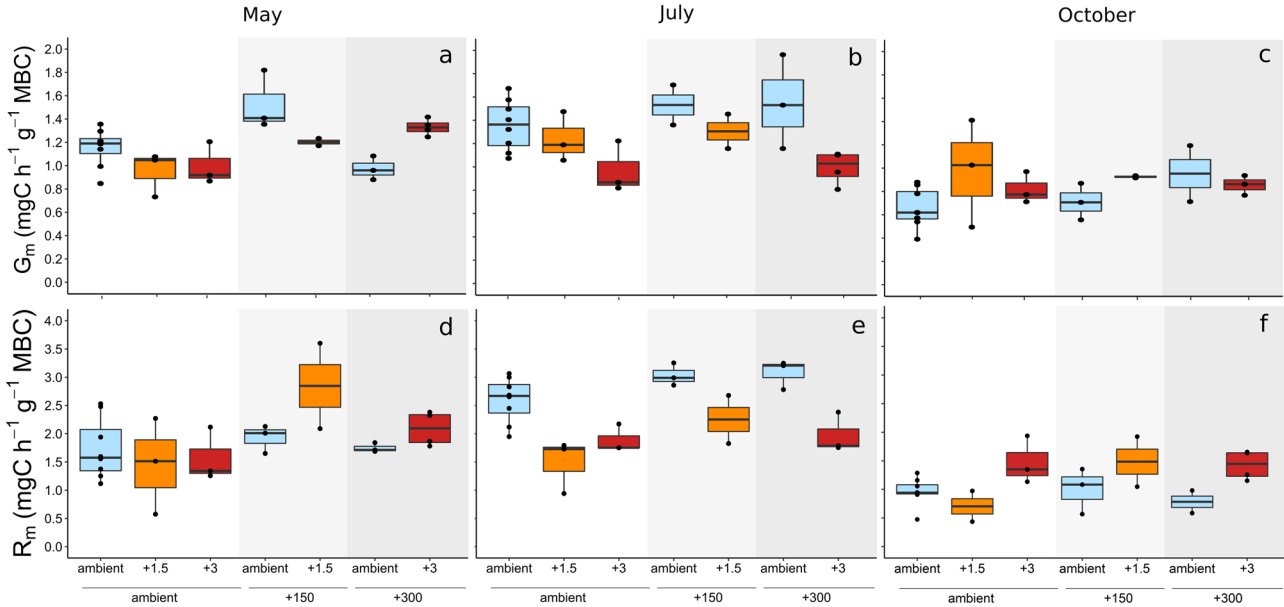

**Fig. 1 Responses of microbial biomass-specific growth and respiration to elevated temperature and atmospheric $CO_2$ concentration.** Microbial biomass-specific growth ($G_m$, mg C $h^{-1}$ $g^{-1}$ MBC) in May (**a**), July (**b**) and October (**c**) and microbial biomass-specific respiration ($R_m$, mg C $h^{-1}$ $g^{-1}$ MBC) in May (**d**), July (**e**) and October (**f**) under various combinations of three temperature - and three $CO_2$ levels: ambient air temperature (ambient, blue), 1.5 °C above ambient temperature (+1.5, orange), 3 °C above ambient air temperature (+3, red); ambient atmospheric $CO_2$ concentration (ambient, white), 150 ppm $CO_2$ above ambient level (+150, light grey), 300 ppm $CO_2$ above ambient (+300, dark grey). MBC microbial biomass carbon. Box centre line represents median, box limits the upper and lower quartiles, whiskers the 1.5x interquartile range, while separated points represents outliers. ($n = 26$ of independent samples in each month, for specific replicate number of each treatment see Methods section, Fig. 5).

reduced soil water content irrespective of climate treatment (current or future climate conditions) (Supplementary Fig. S2).

Microbial biomass carbon (MBC, μg carbon $g^{-1}$ DM) measured by chloroform fumigation, was neither significantly affected by single or combined warming and elevated $CO_2$ in any season (Supplementary Fig. S3 and Table S3), nor affected by summer drought (Supplementary Fig. S4 and Table S4).

**Single and combined effects of warming and elevated $CO_2$ on microbial physiology.** We measured soil microbial community-level growth via ¹⁸O incorporation into microbial DNA and soil respiration over 24 h, which also allowed us to calculate CUE. Warming significantly affected biomass-specific growth ($G_m$, mg C $h^{-1}$ $g^{-1}$ MBC, Fig. 1) and biomass-specific respiration rates ($R_m$, mg C $h^{-1}$ $g^{-1}$ MBC, Fig. 1) at all time points (except for $R_m$ in May; Table 1). The effect was, however, strongly dependent on the sampling date and it interacted with atmospheric $CO_2$. In May, warming alone had a negative effect on biomass-specific growth, whereas in combination with elevated $CO_2$ concentration it promoted biomass-specific growth (Fig. 1a and Table 1). Biomass-specific respiration showed higher values with warming at both $CO_2$ enrichment levels (150 ppm and 300 ppm above ambient atmospheric $CO_2$ concentration) (Fig. 1d) although these differences were not statistically significant (Table 1). In July, we observed lower biomass-specific growth (Fig. 1b) and -respiration rates (Fig. 1e) in warmed plots (Table 1). Reductions of growth and respiration in July occurred irrespective of atmospheric $CO_2$ level (Table 1). Metabolic rates were generally lowest in October, when biomass-specific growth and respiration were again stimulated by warming (Fig. 1c, f and Table 1). At this sampling time point, elevated $CO_2$ had no impact on the magnitude of microbial growth, respiration, or CUE (Table 1).

Over the three sampling time points, measured values of CUE ranged between 0.26 and 0.59 (with average treatment values between 0.32 and 0.56) and was not significantly affected by

season. Microbial CUE did neither significantly respond to combined nor single effects of elevated temperature and atmospheric $CO_2$ enrichment in May and October (Figs. 2a and 3c, and Table 1). In July, we found the highest proportional carbon allocation to growth relative to total uptake at intermediate temperature increase (+1.5 °C) (Fig. 2b). As MBC did not vary across sampling dates, growth, and respiration rates per gram soil (Supplementary Fig. S2) approximately followed the patterns of the biomass-specific rates.

Overall, our models indicated sampling date to be the most significant explanatory factor of variation in biomass-specific microbial growth and respiration. Besides elevated $CO_2$ concentration was identified as another, but less significant predictor of both, growth and respiration rate (Table 2).

**Effects of drought and future climate treatments on microbial physiology.** In plots subjected to a simulated future climate (+3 °C warming in combination with +300 ppm $CO_2$ above ambient conditions) summer drought significantly increased biomass-specific growth and respiration rates (Fig. 3a, b, and Table 3). In contrast, in ambient climate plots, biomass-specific growth was slightly decreased by drought (Fig. 3a), whereas there were no effects on respiration (Fig. 3b).

Two months after the end of drought (terminated with rewetting of the plots mimicking a 40 mm rain event), we found similar values of biomass-specific growth and—respiration rates in previously drought-exposed plots compared to their controls (Fig. 4a, b, and Table 3). Microbial CUE was not affected by drought at any time (Figs. 3c and 4c, and Table 3).

**Discussion**
Most of the available information on the effects of global change factors on ecosystem processes originates from experiments in which single factors were manipulated. However, elevated atmospheric $CO_2$ concentrations, warming, and drought, with

**Table 1 Effect of climate change drivers on microbial biomass-specific growth ($G_m$), biomass-specific respiration ($R_m$) and carbon use efficiency (CUE).**

| | May | | | | July | | | | October | | | |
|---|---|---|---|---|---|---|---|---|---|---|---|---|
| $G_m$ | Est. | SE | t | p | Est. | SE | t | p | Est. | SE | t | p |
| $eCO_2$ | 0.51 | 0.22 | 2.35 | **0.0294** | 0.05 | 0.05 | 1.01 | 0.3229 | 0.01 | 0.04 | 0.15 | 0.8791 |
| eT | −0.56 | 0.18 | −3.04 | **0.0066** | −0.23 | 0.05 | −4.26 | **<0.001** | 0.10 | 0.04 | **2.77** | **0.012** |
| $eCO_2$:eT | 0.12 | 0.04 | 2.81 | **0.0109** | | | | | | | | |
| $eCO_2^2$ | −0.30 | 0.11 | −2.61 | **0.0171** | | | | | | | | |
| $eT^2$ | 0.24 | 0.10 | 2.53 | **0.0202** | | | | | | | | |
| $R_m$ | | | t | p | | | t | p | | | t | p |
| $eCO_2$ | 0.19 | 0.13 | 1.16 | 0.1566 | 0.27 | 0.11 | 2.51 | **0.0200** | 0.01 | 0.09 | 0.16 | 0.8737 |
| eT | 0.06 | 0.13 | 0.45 | 0.6566 | −0.52 | 0.11 | −4.87 | **<0.001** | 0.26 | 0.08 | **3.05** | **0.0062** |
| $eCO_2$:eT | | | | | | | | | | | | |
| $eCO_2^2$ | | | | | | | | | | | | |
| $eT^2$ | | | | | | | | | | | | |
| CUE | | | t | p | | | t | p | | | t | p |
| $eCO_2$ | −0.01 | 0.01 | −0.68 | 0.5034 | −0.03 | 0.08 | −0.41 | 0.685 | 0.01 | 0.02 | 0.444 | 0.6612 |
| eT | 0.00 | 0.01 | −0.23 | 0.8155 | 0.02 | 0.07 | 3.07 | 0.0068 | −0.01 | 0.02 | −0.45 | 0.652 |
| $eCO_2$:eT | | | | | 0.00 | 0.02 | −0.0007 | 0.9994 | | | | |
| $eCO_2^2$ | | | | | 0.01 | 0.04 | 0.29 | 0.7715 | | | | |
| $eT_2$ | | | | | −0.11 | 0.04 | −3.01 | 0.0077 | | | | |

Elevated atmospheric $CO_2$ ($eCO_2$) and increased air temperature (eT) as predictors of biomass-specific growth rate ($G_m$, mgC $h^{-1}$ $g^{-1}$ MBC), biomass-specific respiration rate ($R_m$, mgC $h^{-1}$ $g^{-1}$ MBC) and microbial carbon use efficiency (CUE) at each sampling time point (May, July and October 2017). Values are derived from RSM models. $eCO_2^2$ & $eT^2$ – quadratic functions of elevated $CO_2$ and temperature, $eCO_2$:T interaction of elevated $CO_2$ concentration and temperature. *Est.* estimated slope, *SE* standard error, *p* values < 0.05 are given in bold. (*n* = 26 in each month, for specific replicate number of each treatment see Methods section, Fig. 5).

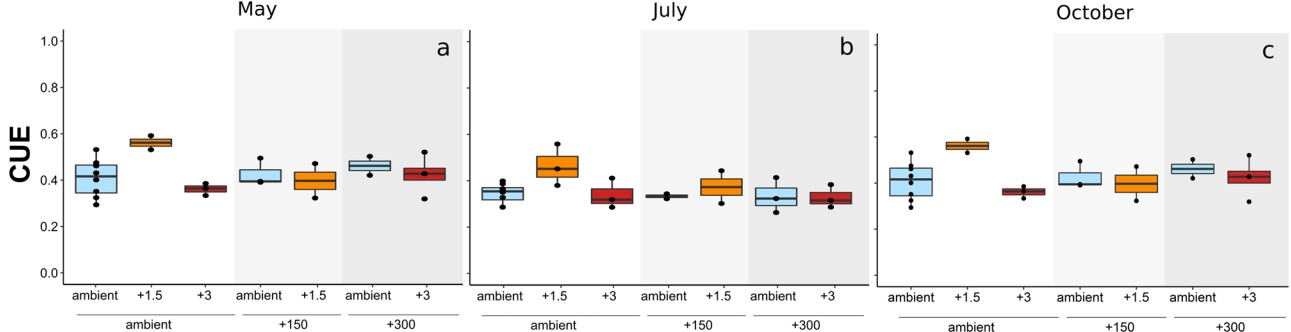

**Fig. 2 Community-level carbon use efficiency under climate change.** Carbon use efficiency (CUE) in May (**a**), July (**b**) and October (**c**) under various combinations of three temperature—and three $CO_2$ levels: ambient air temperature (ambient, blue), 1.5 °C above ambient air temperature (+1.5, orange), 3 °C above ambient air temperature (+3, red); ambient atmospheric $CO_2$ concentration (ambient, white), 150 ppm above ambient levels (+150, light grey), 300 ppm above ambient levels (+300, dark grey). Box centre line represents median, box limits the upper and lower quartiles, whiskers the 1.5× interquartile range, while separated points represents outliers. (*n* = 26 of independent samples in each month, for specific replicate number of each treatment see Methods section, Fig. 5).

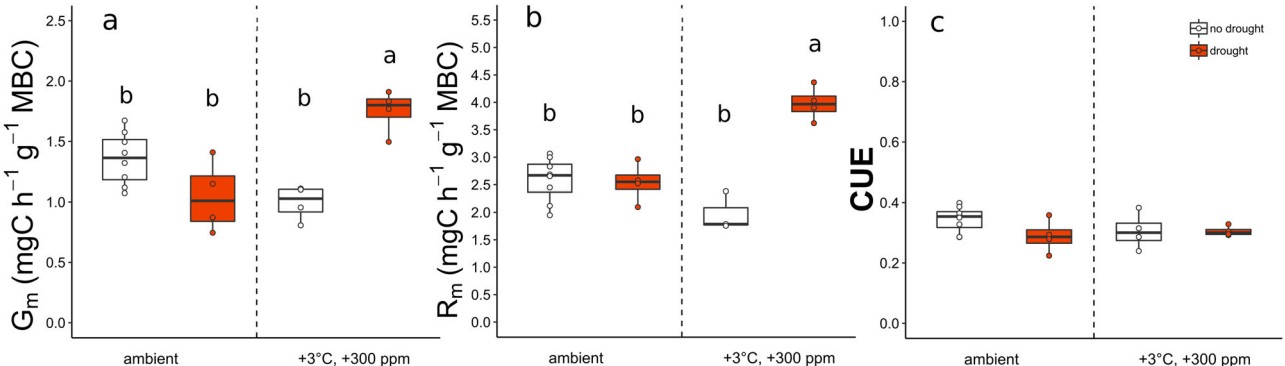

**Fig. 3 Microbial responses to summer drought.** Microbial biomass-specific growth rates ($G_m$, mg C $h^{-1}$ $g^{-1}$ MBC) (**a**), biomass-specific respiration rates ($R_m$, mg C $h^{-1}$ $g^{-1}$ MBC) (**b**) and microbial carbon use efficiency (CUE) (**c**) under ambient precipitation (white) or under rain exclusion (drought, red) at ambient (ambient) or future climate conditions (+3 °C, +300 ppm). MBC microbial biomass carbon. Box centre line represents median, box limits the upper and lower quartiles, whiskers the 1.5× interquartile range. Letters above box-whiskers indicate significant differences between groups (*p* < 0.05, Tukey's HSD test). (*n* = 20 of independent samples in each month, for specific replicate number of each treatment see Methods section, Fig. 5).

**Table 2 Seasonality as a driver of microbial physiology.**

|  | MBC | | | $G_m$ | | | $R_m$ | | | CUE | | |
|---|---|---|---|---|---|---|---|---|---|---|---|---|
|  | df | F | p | df | F | p | df | F | p | df | F | p |
| Date | 1 | 4.24 | **0.0432** | 1 | 18.40 | **0.0001** | 1 | 51.88 | **<0.0001** | 1 | 1.19 | 0.2794 |
| eCO$_2$ | 1 | 0.26 | 0.6134 | 1 | 4.53 | **0.037** | 1 | 4.69 | **0.034** | 1 | 1.15 | 0.2867 |
| eT | 1 | 0.03 | 0.8711 | 1 | 0.46 | 0.4979 | 1 | 1.55 | 0.2171 | 1 | 0.27 | 0.6034 |
| eCO$_2^2$ eT$^2$ |  |  |  | 1 | 8.41 | **0.0051** | 1 | 1.45 | 0.2326 |  |  |  |
| date:eCO$_2$ | 1 | 1.08 | 0.3018 | 1 | 0.13 | 0.7238 | 1 | 0.26 | 0.6101 | 1 | 0.67 | 0.4163 |
| date:eT | 1 | 0.07 | 0.7865 | 1 | 1.57 | 0.2145 | 1 | 3.92 | **0.0106** | 1 | 0.04 | 0.8381 |
| eCO$_2$:eT | 1 | 2.32 | 0.1321 | 1 | 6.72 | **0.0117** | 1 | 0.80 | 0.3746 | 1 | 0.02 | 0.8796 |
| date:eCO$_2^2$ date:eT$^2$ |  |  |  | 1 | 0.37 | 0.5416 | 1 | 0.11 | 0.7442 |  |  |  |
| date:eCO$_2$:eT | 1 | 0.07 | 0.7937 | 1 | 1.62 | 0.2079 | 1 | 0.04 | 0.8388 | 1 | 0.11 | 0.7394 |

Statistical significances of the effect of seasonality (date), elevated CO$_2$ (eCO$_2$) and elevated temperature (eT) on microbial biomass carbon (MBC, µgC g$^{-1}$ DM), biomass-specific growth rate (G$_m$, mgC h$^{-1}$ g$^{-1}$ MBC), microbial biomass-specific respiration (R$_m$, mgC h$^{-1}$ g$^{-1}$ MBC) and microbial carbon use efficiency (CUE). Values are derived from GLS models. eCO$_2^2$ & eT$^2$ represent quadratic functions, ":" indicates the interaction of two or three predictors. df Degree of freedom, p value < 0.05 are given in bold. (n = 26 in each month, for specific replicate number of each treatment see Methods section, Fig. 5).

**Table 3 Biomass-specific growth (G$_m$), biomass-specific respiration (R$_m$) and microbial carbon use efficiency (CUE) during a summer drought and after a 2-month rewetting period.**

|  | df | SS | MS | F | p |
|---|---|---|---|---|---|
| **G$_m$** |  |  |  |  |  |
| Drought |  |  |  |  |  |
| eCO$_2$ + eT | 1.00 | 0.07 | 0.07 | 1.44 | 0.2463 |
| drought | 1.00 | 0.10 | 0.10 | 2.09 | 0.1669 |
| (eCO$_2$ + eT): drought | 1.00 | 1.32 | 1.32 | 27.94 | **<0.0001** |
| Rewetting |  |  |  |  |  |
| eCO$_2$ + eT | 1.00 | 0.21 | 0.21 | 4.81 | **0.04453** |
| drought | 1.00 | 0.00 | 0.00 | 0.03 | 0.86084 |
| (eCO$_2$ + eT): drought | 1.00 | 0.00 | 0.00 | 0.06 | 0.80762 |
| **R$_m$** |  |  |  |  |  |
| Drought |  |  |  |  |  |
| eCO$_2$ + eT | 1.00 | 1.31 | 1.31 | 9.67 | **0.007159** |
| drought | 1.00 | 2.48 | 2.48 | 18.3 | **<0.0001** |
| (eCO$_2$ + eT): drought | 1.00 | 4.42 | 4.42 | 32.64 | **<0.0001** |
| Rewetting |  |  |  |  |  |
| eCO$_2$ + eT | 1.00 | 2.30 | 2.30 | 38.05 | **<0.0001** |
| drought | 1.00 | 0.05 | 0.05 | 0.86 | 0.36667 |
| (eCO$_2$ + eT): drought | 1.00 | 0.38 | 0.38 | 6.3 | **0.02315** |
| **CUE** |  |  |  |  |  |
| Drought |  |  |  |  |  |
| eCO$_2$ + eT | 1.00 | 0.00 | 0.00 | 0.97 | 0.3402 |
| drought | 1.00 | 0.00 | 0.00 | 2.32 | 0.147 |
| (eCO$_2$ + eT): drought | 1.00 | 0.00 | 0.00 | 1.68 | 0.2135 |
| Rewetting |  |  |  |  |  |
| eCO$_2$ + eT | 1.00 | 0.01 | 0.01 | 1.72 | 0.20852 |
| drought | 1.00 | 0.00 | 0.00 | 0.16 | 0.69495 |
| (eCO$_2$ + eT): drought | 1.00 | 0.03 | 0.03 | 0.89 | 0.06619 |

Statistical significances of drought, climate change treatments (eCO$_2$ + eT) and their interaction ((eCO$_2$ + eT): drought) during drought and after a 2-month rewetting period as explanatory variables of biomass-specific growth rate (G$_m$, mgC h$^{-1}$ g$^{-1}$ MBC), biomass-specific respiration rate (R$_m$, mgC h$^{-1}$ g$^{-1}$ MBC) and microbial carbon use efficiency (CUE). Values are derived from two-way ANOVA of each sampling date. df degree of freedom, SS sum of squares, MS Mean Squares, p values < 0.05 are given in bold. (n = 20 in each period, for specific replicate number of each treatment see Methods section, Fig. 5).

high probability, will co-occur[15]. Predictions of ecosystem processes and feedbacks in a future climate therefore are based on the assumption that multiple climate change factors have additive effects[17,18]. This assumption has rarely been tested, and we therefore remain with a poor understanding of how combined climate change factors will affect ecosystems. This valuable information can be derived from multifactorial climate change experiments only. An additional level that has widely been neglected, is how seasonal dynamics directly and indirectly influence the effects of combined climate change factors on soil microbial processes. To our best knowledge, this study is the first to investigate how elevated CO$_2$ concentration, climate warming and episodic drought alone and in combination affect microbial community physiology and how these effects vary seasonally.

In our experiment, season was a better predictor than other global change factors of microbial growth and respiration. Soil microorganisms responded differently to climate change drivers across seasons, suggesting that the main factors underlying the activity of microbial communities in soil were season-specific. Previous studies in grassland systems, also reported a pronounced effect of seasonality on microbial physiology,[36] and on microbial community structure[11]. Indeed, throughout the course of the year soil microbial communities experience a strong variation in many environmental factors, which potentially plays a central role in modulating microbial response to climate change. For example, when temperatures are lowest, the effect of warming is expected to be more pronounced. On the other hand, during summer low soil moisture and decreased plant labile carbon inputs to soil might constrain the responses of microbial physiology, when soil temperature approaches an optimum. However, effects of temperature and plant carbon inputs could not be disentangled in our study.

Warming exerted the most pronounced effects amongst all treatments. Biomass-specific growth and respiration rates increased due to the warming treatment only in autumn, when average microbial activity was the lowest across the year. These findings are consistent with other studies, which indicated that temperature is driving enzyme activity more strongly at low-temperature conditions[51]. With climate warming, microorganisms get closer to the optimal temperature for enzymatic rates[21] and consequently exhibit higher metabolic rates[51]. Indeed, the positive effect of elevated temperature on growth and respiration in October was the opposite of what we observed in July, which is in contrast to previous findings[39,40]. This reversal effect was irrespective of atmospheric CO$_2$ concentration. The negative response of microbial activity to warming in July suggests that other environmental factors might have masked the inherent temperature sensitivity of microbial activity, as previously proposed[44,51]. Particularly, lower soil moisture might have been the leading cause of decreased biomass-specific growth and respiration and might have dampened the positive temperature response. Indeed, we observed a decreased volumetric soil water content in heated plots at all three sampling time points (see Result section: Treatment effects on soil parameters and microbial

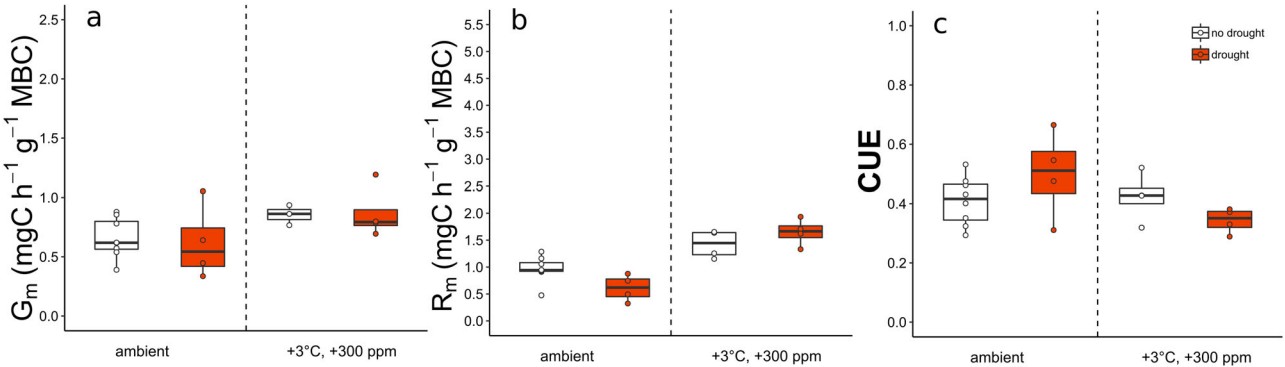

**Fig. 4 Recovery of microbial physiology from drought after 2 months after rewetting.** Microbial biomass-specific growth rate ($G_m$, mg C h$^{-1}$ g$^{-1}$ MBC) (**a**), microbial biomass-specific respiration ($R_m$, mg C h$^{-1}$ g$^{-1}$ MBC) (**b**), and microbial carbon use efficiency (CUE) (**c**) in former to drought-exposed plots (drought) and ambient precipitation plots (white) at ambient (ambient) or future climate conditions (+3 °C, +300 ppm) after a 2-month rewetting period. MBC microbial biomass carbon. Box centre line represents median, box limits the upper and lower quartiles, whiskers the 1.5× interquartile range, while separated points represents outliers. ($n = 20$ of independent samples in each month, for specific replicate number of each treatment see Methods section, Fig. 5).

biomass carbon). Reduction of soil water content in midsummer, when soils were overall driest, might have caused inaccessibility of substrates as the water film within the pore space becomes disrupted[27]. Also other studies indicated that an apparent absence or a negative response of microbial activity to warming was related to low soil water content[52,53] and/or consequently low substrate availability[54]. Factors such as soil moisture and substrate availability were suggested to become restricting if the temperature limitation of metabolic activity is removed[23,51–53]. In spring (May), we observed an interactive effect of warming and elevated $CO_2$ on microbial growth, in contrast to summer and autumn. Specifically, warming alone decreased growth while in combination with elevated $CO_2$ increased it. Plants have been shown to increase belowground carbon input under elevated $CO_2$[13,32,33,35,55–58], including both grasslands and forest ecosystems. Due to this, microbial communities might have depended more on fresh plant carbon input in the early season than in other seasons, which could explain this interactive effect. The interaction between warming and elevated atmospheric $CO_2$ was also observed in other grassland ecosystems[59] and it was argued that plant carbon input into soil exerts a considerable control over the temperature sensitivity of microbial activity[14].

Compared to warming, elevated $CO_2$ concentrations alone exhibited only minor effects on microbial growth and respiration rates. This was expected since elevated $CO_2$ concentrations only affect heterotrophic communities indirectly via effects on plant productivity and soil water content. Plants allocate a bigger portion of carbon belowground to acquire nutrients since elevated $CO_2$ concentrations accelerate the depletion of available nitrogen[13,58]. However, our experiment was conducted in a managed grassland, i.e., a hay meadow that was regularly fertilized. This might have resulted in reduced plant-belowground carbon allocation as would have been expected under elevated $CO_2$ and thus, subsequently masked a possible effect on microbial physiology.

Unexpectedly, drought led to a pronounced acceleration of microbial growth and respiration, but only in the future climate scenario, i.e. in combination with elevated temperature and $CO_2$. The observed increase was not due to differences in soil moisture as ambient climate plots exposed to drought showed a similar soil moisture content (Supplementary Fig. S2E). Increased biomass-specific growth and respiration under drought in warmed plots might have resulted from a shift of the active microbial community towards a community that was better able to cope with reduced soil water content, as it was found in another study[60]. Enhanced rates might also have been caused by increased

belowground plant carbon inputs, although this was not measured. However, care must be taken when interpreting these results. Our method to measure microbial growth depends on adding $^{18}$O-labelled water to the soil. Although we added a relatively small amount of water, that did not strongly alter soil moisture and maintained the differences in soil moisture between drought and ambient treatments, the added water could have caused the changes in microbial growth and respiration. In either case, the response to drought was higher when microorganisms were subjected to a future climate scenario than in controls, which could be due a pre-adaptation of the microbial community to more stressful conditions or indicate higher accumulation of plant-derived substrates in this treatment. Two months after rewetting, we found no lasting effect of drought on microbial growth and respiration, indicating that the microbial community did not experience any legacy effects after the end of the drought period.

While microbial growth and respiration were, to different extents, both affected by warming, elevated atmospheric $CO_2$ concentrations, and drought, we found that microbial CUE was mostly insensitive to any of the treatments (Figs. 2, 4c, and 5c). This observation is consistent with recent warming studies in both grassland[39] and forest system[40]. Although in contrast to theoretical considerations, which expect CUE to decline under warming[5] as respiration is considered more sensitive to temperature increases compared to growth[41]. Our findings demonstrate that the balance of anabolic and catabolic microbial processes was remarkably stable, while both microbial growth and respiration under field conditions are sensitive to warming, elevated $CO_2$, and drought, alone or in combination. This supports the notion that growth and respiration are controlled in a way that the available resources are optimally used under any given condition[6].

To the best of our knowledge this study is the first to investigate changes in soil microbial anabolic and catabolic processes in response to a multifactorial global change manipulation, which integrates seasonal dynamics, enabling us to draw the following conclusions. First, the implementation of a multilevel and multifactorial climate change experiment showed non-linear responses and non-additive interactive effects of climate change on microbial physiology, specifically on growth, but not on microbial carbon use efficiency. The interactive effects between drought and combined elevated $CO_2$ and climate warming are particularly noteworthy, given that they strongly affect our ability to model climate change impacts and microbe-soil-climate

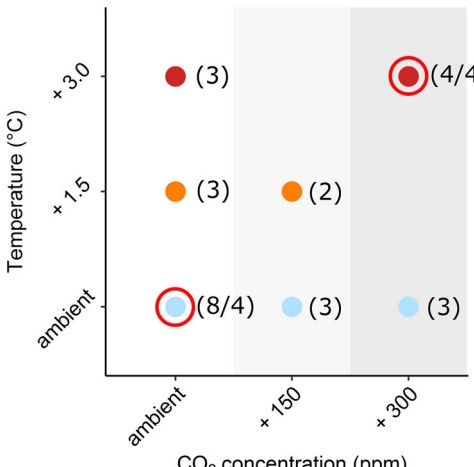

**Fig. 5 Treatment combination and number of replicates used.** The figure illustrates the different combinations of three temperature levels [ambient (light blue), +1.5 °C (orange), +3 °C (red)], three $CO_2$ levels [ambient (white), +150 ppm (light grey), +300 ppm (dark grey)] and drought (red circled dots) used in the experiment. Numbers in brackets represent the number of replicate plots per treatment, number after the slash refers to the available replicates for the drought treatment.

feedbacks. Second, while seasonal dynamics have been shown to modulate climate change effects on biological systems, several studies fail to account for these dynamics in their experimental design. From our study we conclude that seasonal changes in a managed grassland ecosystem (e.g., temperature, soil moisture, or plant carbon inputs), played an important role in shaping the responses of microbial growth and respiration to global changes. Finally, although the individual and combined effects of climate change treatments on microbial growth and respiration rates were significant, we found community-level microbial CUE to be remarkably robust, irrespective of treatment or seasons. This suggests that future research should focus on microbial growth instead of CUE alone, in order to understand and predict the effects of global change on soil carbon dynamics.

## Methods

**Field site and soil sampling**. This study was conducted within the scope of a multifactorial climate change experiment (named ClimGrass) at the Agricultural Research and Education Centre (AREC) in Raumberg-Gumpenstein. The study site is located in Styria, Austria (47°29′37″N, 14°06′0″E), 710 m above sea level and was established in a managed sub-montane grassland. Plots were sown with a local mixture of plant species adopted for the establishment of permanent grasslands ("Dauerwiese B" of the AREC) in 2007. The soil is classified as Cambisol with loamy texture[61] with a pH of 5 (in 10 mM $CaCl_2$) in the upper 10 cm. Aboveground biomass is mown and removed three times a year. Plots are regularly amended with mineral fertilizer replacing the amount of nutrients removed by the harvests (spring: 30 kg N, 32.5 kg P, 85 kg K, after first harvest: 30 kg N, after second harvest: 30 kg N).

The experimental design (the number of replicates per treatment) was based on a response surface regression approach for the warming and $CO_2$ treatments[50] and was combined with a factorial design testing for drought effects under ambient versus future (+3 °C warming, +300 ppm $CO_2$) conditions. The experiment comprises in total 54 plots (4 × 4 m each) showing various combinations of three different levels of temperature (ambient, +1.5 °C, +3 °C) and atmospheric $CO_2$ concentration (ambient, +150 ppm, +300 ppm), and is provided with automated rain-out shelters to simulate summer drought (only on specific plots, which are circled in red in Fig. 5) and Supplementary Table S1). From the total of 54 plots we chose 34 for this study (Fig. 5 and Supplementary Table S1).

Potential future climate scenarios are simulated through the combination of infrared heating systems to increase air temperature and a mini-FACE (Free Air $CO_2$ Enrichment) approach for fumigation with $CO_2$ since May 2014. Infrared heaters are switched on all year round except when snow cover exceeds 10 cm. $CO_2$ fumigation is only applied during daytime throughout the growing season (beginning of April until the end of November) when radiation energy exceeded

50 W m$^{-2}$. Plots that were not heated or fumigated were equipped with not-functional heaters and/or miniFACE rings of the same shape and size to account for possible disturbances.

In 2017 (the fourth year of operating) automated rain-out shelters which were installed above four ambient climate plots (ambient $CO_2$ concentration and ambient temperature) and four future climate plots (receiving a combination of +3 °C and +300 ppm above ambient levels; Fig. 5) were activated. Automatic rain-out shelters were operated for two months, starting from the 23rd of May 2017 until the 27th of July 2017. Rain exclusion was performed in two phases: a pre-conditioning period between May 23rd and June 26th, during which soil moisture was progressively reduced, but five small natural rain events, amounting to a total of 32 mm, were permitted. This was followed by a complete rain exclusion period between June 26th and July 27th, during which the shelters were automatically closed at the onset of all rain events. After the second harvest plots were rewetted with 40 mm of collected rainwater (July 27th). This amount represents a typical heavy precipitation event in the studied area and provides sufficient water for achieving a complete and homogeneous rewetting without incurring in surface run off. In parallel to rewetting, rain-out shelters were deactivated.

We collected soil samples directly after the aboveground biomass had been cut at three time points during the growing season in 2017: in spring (30th and 31st of May), midsummer (25th and 26th of July) and beginning of autumn (3rd and 4th of October). For the drought treatment these dates represent the onset of drought, peak drought before rewetting and the recovery period, respectively. Three to eleven soil cores of 2 cm in diameter were taken from the upper 10 cm of the soil profile in the centre of the 34 plots 1 h after aboveground biomass was harvested. Soil cores were pooled to obtain one composite sample per plot. We removed stones, roots, and shoot residues from soil samples by sieving to 2 mm directly after soil cores were taken. All experiments and laboratory assays were performed at the respective field temperatures measured at the time of harvest (Table S2).

**Soil parameters and microbial biomass carbon**. The volumetric soil water content (SWC) was measured at 1-min intervals with soil moisture sensors (SM150T, DeltaT) and recorded as 15 min averages. Sensors were inserted at 3- and 9-cm depth and operated throughout the entire growing season in a subset of plots representative of all treatments. We averaged the two depth measurements to depict changes in topsoil water content. Soil water content in the collected soil samples was determined gravimetrically by weighing 5 g of the fresh soil and drying at 95 °C for 24 h. This was done the day before the start of the incubation experiment in order to calculate how much $^{18}$O-labelled water or DNAse-free water could be added to the soils and was repeated a second time before soil amendment to determine the precise soil water content of the incubated samples (used for calculations of enrichment of the total soil water). Soil pH was determined in a 1:5 (w:v) mix of fresh soil and 0.01 M $CaCl_2$ solution with a pH meter (Sentron, The Netherlands).

Microbial biomass carbon (MBC) was measured via the chloroform-fumigation extraction (CFE) method[62]. Briefly, one subset of samples was directly extracted in 1 M KCl (in a 1:7.5 w:v ratio of extractant to soil) for 30 min and filtered through ash-free filters, representing the extractable organic carbon (EOC). The other subset was extracted after 24 h of chloroform fumigation (started at the same day as the soil amendment with $^{18}$O-labelled water, see following section). The fumigated samples were extracted in the same way as the non-fumigated samples. The extracts were stored at −20 °C until analysis of extractable organic carbon (EOC) on a TOC/N Analyzer (TOC- VCPH/CPNT-NM-1, Shimadzu, Japan). Microbial biomass carbon was calculated as the difference between EOC in the fumigated sample minus EOC in the non-fumigated sample using a correction factor of 0.45[62].

**Microbial physiology metrics**. In order to understand the effects of the different treatments on microbial physiology, we measured microbial growth and respiration and calculated community-level CUE. We estimated growth rates and CUE of the microbial communities by a substrate-independent method, i.e., via the incorporation of $^{18}$O from labelled soil water into DNA[63]. For this, two subsets of 400 mg field moist soil of each sample were weighed into 1.2-ml cryovials. The open cryovials were placed in headspace glass vials (27 ml), which were then sealed air-tight with a rubber seal. One subset of soil samples was amended with $^{18}$O-labelled water (Campro Scientific) of various $^{18}$O enrichments, in order to reach approximately 25 at% of $^{18}$O in the final soil solution and to concurrently maintain differences in the soil water content of samples. Natural $^{18}$O abundance, i.e. control samples were amended with the same volume of DNAse-free water instead of $^{18}$O-labelled water. After the amendment, vials were incubated for 24 h at their respective treatment temperature (Supplementary Table 2). After the incubation, the cryovials were closed, frozen in liquid nitrogen and stored at −80 °C. DNA was extracted from the entire sample from labelled soil samples and natural abundance controls using a DNA extraction kit (FastDNA$^{TM}$ SPIN Kit for Soil, MO Biomedicals). Extractions were carried out according to the manufacturer's instructions, with two exceptions: the initial centrifugation step was extended to 15 min in order to gain a larger proportion of the cell debris from the supernatant, and the entire matrix containing the DNA was loaded on the SPIN$^{TM}$ filter. DNA extracts were stored at −80 °C. The concentration of dsDNA was measured fluorimetrically using the PicoGreen$^{®}$ Assay (Quant-iT$^{TM}$ PicoGreen$^{®}$ dsDNA Reagent, Life

Technologies). To determine the isotopic ratio of $^{18}O$ to $^{16}O$ of the DNA, 50 μl aliquots of the DNA extracts were dried in silver capsules for 24 h at 60 °C to remove all water. The $^{18}O$ abundance (at% $^{18}O$) and total oxygen content of soil DNA samples were then measured using a Thermochemical Elemental Analyser (TC/EA Thermo Fisher) coupled via a Conflo III open split system (Thermo Fisher) to an Isotope Ratio Mass Spectrometer (IRMS, Delta V Advantage, Thermo Fisher).

To assess microbial respiration during the incubation period, 5 ml gas samples were taken at two timepoints from each headspace vial (one directly after water amendment to the soil and the other at the end of the incubation) and transferred to pre-evacuated 3 ml exetainer vials. Air removed during the first gas sampling was replaced with 5 ml of air with known $CO_2$ concentration. $CO_2$ concentrations were determined by gas chromatography (Trace GC Ultra, Thermo Fisher Scientific, Vienna, Austria) equipped with a vacuum dosing system (S + H Analytics, Germany) and a flame ionization detector (FID) with a methanizer for $CO_2$. Microbial respiration rates (R) were expressed as the amount of $CO_2$ being produced per hour and gram soil dry mass during the 24-h incubations.

To determine microbial community growth (G; expressed as µg carbon per hour per gram of soil dry mass) and microbial carbon use efficiency (CUE), we calculated the amount of DNA produced during the incubation period ($DNA_p$; expressed as µg DNA per hour per gram of soil dry mass). The production of new DNA was calculated as the difference in $^{18}O$ abundance between the labelled and the natural abundance samples using a factor of 31.21, which describes the proportional mass of O content (weight%) in an average DNA formula. Based on soil DNA concentration and microbial biomass carbon that we determined for each sample, we calculated growth as

$$G = DNA_p \; x \; \frac{MBC}{DNA} \qquad (1)$$

In order to obtain microbial community CUE, we divided microbial growth by total carbon uptake (U; expressed as µg carbon per hour per gram of soil dry mass), which was calculated as the sum of microbial growth and respiration:

$$CUE = \frac{G}{U} = \frac{G}{G+R} \qquad (2)$$

**Statistics and reproducibility**. We conducted all statistical analyses and graphs in R (3.4.2). The significance threshold was set to 0.05 for all statistical tests. Data were statistically analysed following two statistical models (response surface model and ANOVA model; Supplementary Table 1).

**Response surface model**. To test the effects of season, warming, and $CO_2$ concentration and their interaction on microbial physiology, we built Generalised Least Square (GLS) regression models using the function *gls* of the R package "nlme"[64]. We built different models varying in complexity. In all models, the three levels of temperature and atmospheric $CO_2$ were considered as numeric fixed factors, in both linear and quadratic terms, as well as their linear-by-linear interaction. Sampling date was added to the model also in its interaction with the linear term of warming and elevated $CO_2$. Because sampling dates resulted in large differences in mean variance, we included the *varIdent* function to the *weights* argument to allow for heterogeneous variance between dates[65]. To account for potential autocorrelation between sampling dates, we integrated different autocorrelation corrections (functions *corCAR1, corAR1, corSymm, corCompSymm*) in our selected models. Non-significant terms were dropped following the marginality principle[66]. When the interaction term of both covariates was significant, the two linear main effects were kept. To assess the presence of possible autocorrelation, we fitted an autocorrelation function (ACF) and inspected the resulting plots. ACF is estimated by calculating the correlation between pairs of log-transformed population densities, between time lags in the feedback response. The autocorrelation coefficients were then plotted against the lags to give the ACF. ACF reveals periodic patterns more clearly than the time plot and also provides an objective estimate of the cycle period[67–71]. When the best-fitting model was chosen, we checked for homogeneity of variances and normality of residuals as previously suggested[72] by inspecting plot of standardized residuals versus predicted values, frequency histogram and QQ-plot. *P* values of the chosen model were generated using the function *anova*. Microbial biomass-specific growth rate, respiration rate and biomass-specific respiration rate were log-transformed to meet the assumption of normality and homogeneity of variances. In order to determine how microbial growth, respiration and community-level CUE were affected by climate change drivers within a specific season, we fitted multiple response surface models (RSMs) with increasing complexity, using the function *rsm* of the R package "rsm"[73].

The *rsm* function automatically generates a lack-of-fit test (to examine the overall model performance by means of R2 and *p* value) and allows to assess the significance of each term added to the model (individual linear factors, two-way interaction and quadratic terms). Based on the output, non-significant terms were dropped, following the marginality principle[66]. Once the best fitting model was chosen, we checked for homogeneity of variances and normality of residuals (see above).

In the results section we display the output of the *anova* function of the chosen model, which shows the significances of explanatory factors by displaying the *t* and *p* values.

The number of replicates in each season was 26, with each replicate representing an individual plot. The number of replicates varied across treatments (for replicate numbers see Fig. 5 in the Material and Methods section or Table S1 in the Supplementary Material).

**Anova**. To test for the effect of drought on microbial physiology in ambient climate and future climate plots (+3 °C, +300 ppm), we performed a two-way ANOVA including climate treatment, drought and their interaction as main factors using the *aov* function. This was done for each individual date separately. Subsequently, we checked for homogeneity of variances, normality of residuals and potential outliers. If all assumptions were met, we performed a Tukey's HSD (function *TukeyHSD*) as post-hoc test for each date to check for significant differences between treatments.

For this analysis we had a sample size of 20 replicates for each time point. Each replicate represented an individual treatment plot. The number of replicates varied across treatments (for replicate numbers see Fig. 5 in the Material and Methods section or Table S1 in the Supplementary Material).

**Reporting summary**. Further information on research design is available in the Nature Research Reporting Summary linked to this article.

## Data availability
The authors declare that the data supporting the findings of this study are available within the Supplementary information files, under the name Supplementary Data 1.

## Code availability
The codes created to analyze the datasets during the current study are available from the corresponding authors on reasonable request.

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

## Acknowledgements

We thank Margarete Watzka and Ludwig Seidl for assistance in the laboratory. This work was supported by the Austrian Science Fund FWF [grant number P 28572] and the Austrian Ministry of Agriculture, Regions, and Tourism. We want to thank the Austrian Research and Education Centre Raumberg-Gumpenstein (AREC) for assisting us during the sampling campaigns and providing the experimental site, which was supported by the DaFNE project ClimGrassEco (101067).

## Author contributions

A.R., W.V., and M.B. conceived the original idea. E.S., V.M., J.S., and T.B. carried out the experiment, and A.C., A.R., W.V., M.B., and E.P. supervised the project. H.P. developed the statistical approach and DR contributed with field site management and soil moisture data. E.S., A.C., and A.R. wrote the original draft and all authors reviewed the manuscript.

## Competing interests

The authors declare no competing interests.
