## [Peer Review File · Communications Biology]

Reviewers' comments:

Reviewer #1 (Remarks to the Author):

This is a study on the effects of temp, CO₂, season and drought on soil microbial growth rates, respiration and CUE as measured by ¹⁸O water incorporation into microbial DNA. It is well written and the research is well conducted. I recommend acceptance.

I have a few small points that the authors may wish to address:

L46: it is not correct to associated respiration with energy production: There are many reactions that release CO₂ and that do not involve energy production. For example, in the lysine production, CO₂ is released during biosynthesis but not associated with ATP or NAD(P)H production. Moreover, CUE is a complex character with efficiency of biochemical processes as only one factor. Other factors include turnover, exudation and others (Manzoni et al 2012, Hagerty et al 2018, Geyer et al 2016, 2019). This is not reflected in the introduction, instead CUE is described as the result of microbial physiology and energy metabolism only.

L102: it may be good to mention in the text the temperatures and the CO₂ concentrations. Also, it could be made clear better how drought treatment was applied in the text, and not only in the method section at the end.

L130: the word both is surprising if there are three levels of CO₂ – this only becomes clear at the end or in the supplement – this should be made clear earlier.

L149: it is not clear to me whether the range of CUE values is based on treatment averages (which I think it should be) or on individual datapoints (which is less interesting). Please clarify.

L167: remove "future" or "change treatment". It is unlikely that microbes will respond to future treatments

L172: an, when reading from top to bottom, unexpected treatment – please clarify the rewetting treatment. Why was 40 mm chosen?

The discussion could be streamlined a bit, and perhaps, if allowed, divided into sections with clear headings

L279: please explain theoretical considerations.

L310: what is HBLFA?

L382: is this 400 mg field moist soil?

L405: why is it "around" 200 ppm?

Supplemental figure S2: is the placement of vertical lines OK. It seems that the moisture content changes after launching and before removing rain shelters

Legend with supplemental figure S5: text legend incomplete?

Reviewer #2 (Remarks to the Author):

I would like to thank the editor for inviting me to review this paper. The authors investigated the

microbial growth, respiration, and carbon use efficiency of a managed grassland under the combination of three levels of warming, CO₂, and summer drought. The outcomes provide vital knowledge for understanding soil physiological processes in response to future climate change. The stable CUE also indicated a strategy of microbes to maximize energy use under unfavorable conditions. As I am not an expert in the microbial fields, the following suggestions are based on my knowledge.

1. Abstract. Some numbers from your results should be added rather than just some descriptive sentences of the results;

2. Introduction. The experiment was carried out in a managed grassland ecosystem. Are there more related research in forests than in grassland? I suggest the authors add the research advances of microbial growth and CUE in grassland ecosystems;

3. Results. Line 114-122. This section can be combined in the Discussion section to support your main results;

4. Line 156-157. More information about Table 2 should be added, such as the ECO₂ also significantly affect the G_m and R_m.

5. Discussion. The authors discussed the impacts of season, warming, enhancement of CO₂, and drought on microbial growth and respiration. It would be clearer if the subdivisions can be provided.

The methods are quite clear and robust in this paper. Given the significance and overall quality of this paper, I think it can be accepted after these revisions.

We have received two reports on your manuscript. I agree the review #3 that you should improve the extend explanation of your results rather than just a description in the revised version.

We thank the editor for giving us the possibility to answer the points raised by the referees. We have answered each comment here below (answers are in blue) and we have detailed the exact lines where the modifications were made. We have also highlighted in yellow each modification on the revised version of the manuscript to make the revision process easier. We believe that we addressed all the comments and that we have now extended the explanation of the results.

Referee expertise:

Referee #1: microbial-ecology and global change ecology

Referee #2: global change ecology

Reviewers' comments:

Reviewer #1 (Remarks to the Author):

This is a study on the effects of temp, CO₂, season and drought on soil microbial growth rates, respiration and CUE as measured by ¹⁸O water incorporation into microbial DNA. It is well written and the research is well conducted. I recommend acceptance.

We thank the reviewer for the positive comment and for the time taken to improve the manuscript. We address each individual comment here below.

I have a few small points that the authors may wish to address:

L46: it is not correct to associated respiration with energy production: There are many reactions that release CO₂ and that do not involve energy production. For example, in the lysine production, CO₂ is released during biosynthesis but not associated with ATP or NAD(P)H production. Moreover, CUE is a complex character with efficiency of biochemical processes as only one factor. Other factors include turnover, exudation and others (Manzoni et al 2012, Hagerty et al 2018, Geyer et al 2016, 2019). This is not reflected in the introduction, instead CUE is described as the result of microbial physiology and energy metabolism only.

We agree with the reviewer's comment that CO₂ production cannot directly be equated with energy production and that microbial CUE is more complex than the simplification we provided in the introduction. We changed the wording of this section and added more information in the introduction to offer a more comprehensive explanation of microbial CUE and the methodological limitation involved, while trying to still be concise (see L. 47-53 of the updated version)

L102: it may be good to mention in the text the temperatures and the CO₂ concentrations. Also, it could be made clear better how drought treatment was applied in the text, and not only in the method section at the end.

We added the three temperature and CO₂ levels in brackets in the text (Line 106-108) and added a sentence describing the drought treatment (Line 108-112).

L130: the word both is surprising if there are three levels of CO₂ – this only becomes clear at the end or in the supplement – this should be made clear earlier.

We amended the text to clarify that we referred only to the two elevated CO₂ levels (now in Line 136, 137).

L149: it is not clear to me whether the range of CUE values is based on treatment averages (which I think it should be) or on individual datapoints (which is less interesting). Please clarify.

The numbers reflected the range of individual CUE values across the three harvests, as we wanted to display the full range of measured values; we clarified that and additionally added treatment averages of CUE (now Lines 157, 158).

L167: remove “future” or “change treatment”. It is unlikely that microbes will respond to future treatments

We thank the reviewer for pointing this out. We now removed the word “change” to make the sentence clearer (now Line 180).

L172: an, when reading from top to bottom, unexpected treatment – please clarify the rewetting treatment. Why was 40 mm chosen?

We added a line about the end of the drought treatment in the Results section (Line 185, 186).

The 40 mm event was chosen for two main reasons: 1) during the summer of 2017 there were several single rain events around 40mm (the night before the rewetting the field experienced a single rain event of 45,2 mm) and therefore 40 mm is representative of a single rain event in the studied area; 2) previous tests at the site showed that 40 mm was the maximum amount that can be applied during a 12 hour period to dried soil without causing surface runoff. We now added these details in the materials and methods section (Line 366-369).

The discussion could be streamlined a bit, and perhaps, if allowed, divided into sections with clear headings

We agree with the reviewer that subheadings would make the discussion clearer.

Unfortunately the journal guidelines state that the discussion section may not contain subheadings (see <https://www.nature.com/commsbio/submit/content-types>).

L279: please explain theoretical considerations.

Theoretical studies suggest that respiration is more sensitive to increasing temperatures than growth, which would yield decreasing CUE at increasing temperatures. We added this explanation to the discussion (now Line 297-299).

L310: what is HBLFA?

We apologize for not having explained what HBLFA means. It is the German abbreviation for Agricultural Research and Education Centre; we have corrected this now (now Line 327, 330) .

L382: is this 400 mg field moist soil?

Yes, we now clarified this in the text (now Line 407).

L405: why is it “around” 200 ppm?

The reason of the word “around” is that the air used did not always have exactly 200 ppm (the gas bottle was changed between the samplings). However, the precise CO₂ concentration was always measured and used in the calculations. To avoid confusion for the reader, we decided to delete this information (deleted the words “at around 200 ppm”) as is not necessary to understand the procedure.

Supplemental figure S2: is the placement of vertical lines OK. It seems that the moisture content changes after launching and before removing rain shelters

We thank the reviewer for noticing the mistake. The second vertical line (end of the drought period) was misplaced and corrected (Supplementary figure S2). The first vertical line placement was correct. The reason why soil moisture content increases after the beginning of the operation of the rainout shelters is that the rain exclusion was performed in two phases: a pre-conditioning period between May 23rd and June 26th, during which soil moisture was progressively reduced, but five small natural rain events, amounting to a total of 32 mm, were permitted. This was followed by a complete rain exclusion period between June 26th and July 27th, during which the shelters were automatically closed at the onset of all rain events. We now added this explanation in the Methods section (now Line 362-366). We also attached the updated figure here below:

Legend with supplemental figure S5: text legend incomplete?

We thank the reviewer for noticing the mistake. The text window somehow changed when the figure was converted to pdf. We now corrected the error.

Reviewer #2 (Remarks to the Author):

I would like to thank the editor for inviting me to review this paper. The authors investigated the microbial growth, respiration, and carbon use efficiency of a managed grassland under the combination of three levels of warming, CO₂, and summer drought. The outcomes provide vital knowledge for understanding soil physiological processes in response to future climate change. The stable CUE also indicated a strategy of microbes to maximize energy use under unfavorable conditions. As I am not an expert in the microbial fields, the following suggestions are based on my knowledge.

We thank the reviewer for the positive comments and for the time taken to review the manuscript. We tried to include all the suggestions proposed by the reviewer or justify in case we could not include them.

1. Abstract. Some numbers from your results should be added rather than just some descriptive sentences of the results;

We added the average CUE value in parentheses (now Line 28) to address the reviewer suggestion. We were unable to add more numeric results in the Abstract due to word number restrictions.

2. Introduction. The experiment was carried out in a managed grassland ecosystem. Are there more related research in forests than in grassland? I suggest the authors add the research advances of microbial growth and CUE in grassland ecosystems;

We appreciate the reviewer's comment. As mentioned in the introduction, current literature on microbial growth and CUE affected by climate change is limited. Therefore, in order to address the reviewer's comment, we implemented 2 things:

- 1) We specified literature belonging to grassland ecosystems throughout the manuscript (e.g. Line 86-87 and Line 260);
- 2) We added specific statements on the latest advances on microbial physiology and CUE in grasslands in the discussion section (Line 228-229, Line 262-263 and Line 297-298).

We did not add specific statements in the Introduction as we aimed at maintaining it general given that only at the end of the introduction the system under study is specified.

3. Results. Line 114-122. This section can be combined in the Discussion section to support your main results;

We agree with the reviewer that this part of the result section is useful in the discussion of our main result. We were unable, however, to completely combine this part of the Results section with the Discussion, in order to keep the two sections separated. We have tried instead to refresh these arguments in the Discussion again and make the link to the result section (Lines 250-251).

4. Line 156-157. More information about Table 2 should be added, such as the eCO₂ also significantly affect the G_m and R_m.

We added a sentence on the significance of eCO₂ as a predictor of biomass-specific growth and respiration (Line 166, 167), to address the reviewer's suggestion.

5. Discussion. The authors discussed the impacts of season, warming, enhancement of CO₂,

and drought on microbial growth and respiration. It would be clearer if the subdivisions can be provided.

We agree with the reviewer that subheadings could make the discussion clearer, which also was a comment from Reviewer 1. Unfortunately the journal guidelines state that the Discussion section may not contain subheadings (see: <https://www.nature.com/commsbio/submit/content-types>).

The methods are quite clear and robust in this paper. Given the significance and overall quality of this paper, I think it can be accepted after these revisions.

REVIEWERS' COMMENTS:

Reviewer #2 (Remarks to the Author):

The authors have addressed the comments properly, and made corrections. Therefore, I suggest this MS can be accepted for publication.